# THC and sperm: Impact on fertilization capability, pre-implantation in vitro development and epigenetic modifications

Alexander G. Kuzma-Hunt[1], Reem Sabry[1], Ola S. Davis[1], Vivien B. Truong[1], Jibran Y. Khokhar[2], Laura A. Favetta[1]*

1 Reproductive Health and Biotechnology Lab, Department of Biomedical Sciences, University of Guelph, Guelph, Ontario, Canada, 2 Department of Anatomy and Cell Biology, Western University, London, Ontario, Canada

* lfavetta@uoguelph.ca

**Data Availability Statement:** All relevant data are within the manuscript and its Supporting information files.

## Abstract

Global cannabis use has risen 23% since 2010, with 209 million reported users, most of whom are males of reproductive age. Delta-9-tetrahydrocannabinol (THC), the main psychoactive phytocannabinoid in cannabis, disrupts pro-homeostatic functions of the endocannabinoid system (ECS) within the male reproductive system. The ECS is highly involved in regulating morpho-functional and intrinsic sperm features that are required for fertilization and pre-implantation embryo development. Previous work by our group demonstrated that THC altered sperm capacitation and the transcriptome, including several fertility-associated microRNAs (miRs). Despite the prevalent use of cannabis among males of reproductive age, clinical and pre-clinical research investigating the impact of paternal cannabis on sperm function and the outcomes of artificial reproductive technologies (ARTs) remains inconclusive. Therefore, the present study investigates the impact of in vitro THC exposure on morpho-functional and intrinsic sperm functions, including contributions to embryo development following IVF. Bovine sperm were used as a translational model for human and treated with concentrations of THC that reflect plasma levels after therapeutic (0.032μM), and low (0.32μM)-high (4.8μM) recreational cannabis use. After 6-hours of treatment, THC did not alter the acrosomal reaction, but 4.8μM significantly reduced mitochondrial membrane potential (MMP) (p<0.05), primarily through agonistic interactions with CB-receptors. Fertilization of bovine oocytes with THC-treated sperm did not alter developmental rates, but blastocysts generated from sperm treated with 0.32–4.8μM THC had fewer trophoblasts (p<0.05), while blastocysts generated from sperm exposed to any concentration of THC had fewer cells in the inner cell mass (ICM), particularly within the 0.032μM group (p<0.001). Fertility associated miRs, including miR-346, miR-324, miR-33b, and miR-34c were analyzed in THC-exposed sperm and associated blastocysts generated by IVF, with lower levels of miRs-346, -324, and -33b found in sperm treated with 0.32μM THC, while miR-34c levels were higher in sperm treated with 0.032μM THC (p<0.05). Levels of miR-346 were also lower in sperm treated with 0.032μM THC, but higher in blastocysts generated from sperm exposed to 0.32μM THC (p<0.05). Our findings suggest that THC may alter key morpho-functional and epigenetic sperm factors involved in fertilization and embryo

**Funding:** The author(s) received no specific funding for this work.

**Competing interests:** The authors have declared that no competing interests exist.

development. This is the first study to demonstrate that sperm exposed to THC in vitro negatively affects embryo quality following IVF.

## Introduction

In 2023, the global prevalence of lifetime and 12-month-window infertility was estimated to be 17.5% and 12.6%, respectively, affecting approximately one in six people [1]. Infertility is a universal healthcare challenge, with similar rates across countries of all economic statuses [1]. Male infertility plays a role in approximately 50% of infertility cases, being the sole cause in 30% and contributing another 20% within the couple [2]. Instances of male infertility are expected to rise given that global sperm counts declined by 51.6% from 1973 to 2018 and continue to do so at an accelerated rate [3]. Although in vitro fertilization (IVF) helps many to achieve pregnancy, recent data suggests that live births per fresh IVF/intracytoplasmic sperm injection (ICSI) cycle in the US substantially declined between 2004 to 2016 [4, 5]. Biological factors cannot solely explain rising rates of infertility and failed IVF cycles, shifting the focus of research towards a more inclusive view of male reproductive health that focuses on lifestyle and environmental factors such as alcohol use, obesity, endocrine disrupting compounds, and cannabis use [1–3, 6–8].

Globally, the number of cannabis users aged 15–64 rose 23% from 170 million in 2010 to 209 million as of 2022 [9, 10]. The 2022 Canadian Cannabis Survey indicated that 27% of Canadians aged 16 and older used cannabis in the past 12 months [10]. Most Canadian cannabis users are of reproductive age, with more males reporting daily usage compared to females [10, 11]. Additionally, cannabis use disorder is more common in males and increased from 4.9% in 2014 to 5.9% in 2018 among US adults 18–25 years old [12–14].

Despite the prevalence of cannabis use among adults of reproductive age, there are no clear clinical recommendations regarding cannabis and male fertility [15]. Jordan et al. (2020) reports that cannabis use among infertility patients was similar to the general population, and that only 9.4% of patients who reported cannabis use to their physician were advised to stop [15]. The lack of clear medical guidelines regarding cannabis use and male fertility is largely a result of inconclusive research. Some studies report a harmful effect of cannabis on sperm counts, morphology, motility, capacitation and the acrosomal reaction (AR) [16–22], while others have associated cannabis use with protection against abnormal sperm motility [16] and higher sperm counts [23]. Despite the harmful impact of cannabis on sperm [21], there is very limited research investigating the effects of paternal cannabis use on more clinically relevant outcomes, such as embryo quality following IVF [24, 25]. Clinical data concerning the impacts of paternal cannabis on reproductive function have been limited to observational studies and self-reported cannabis use [24, 25].

Reproductive health risks associated with cannabis use are predominantly attributed to delta-9-tetrahydrocannabinol (THC) [26]. THC interacts with the endocannabinoid system (ECS) as a partial agonist at cannabinoid receptors 1 and 2 (CB1, CB2), and non-classical ECS receptors [26–28]. In vivo mice studies show that THC exerts both agonist and antagonist effects following acute administration, as THC would antagonize the hypothermic effects of another cannabinoid agonist [29]. CB1 and CB2 are transmembrane G-protein-coupled receptors (GPRs) that initiate multiple intracellular events including: 1) the activation of kinases such as mitogen-activated protein kinase (MAPK) and extracellular regulated kinases (ERKs); 2) inhibition of soluble adenyl cyclase, subsequently reducing cyclic adenosine

monophosphate (cAMP) and protein kinase A (PKA) activation, and 3) changes to intracellular calcium and potassium levels [30–33]. ECS signalling is involved in essential male reproductive functions including steroidogenesis, spermatogenesis, and sperm function [34–36].

ECS signalling via endogenous cannabinoids (ECBs), anandamide (AEA) and 2-arachidonolyglyercol (2-AG), regulates several morpho-functional sperm features including mitochondrial activity, the AR, and capacitation [21, 34, 35, 37–39]. Most evidence indicates that aberrant ECS signalling negatively impacts mature sperm function. The seminal plasma of men with asthenozoospermia and oligoasthenoteratozoospermia has lower ECB levels [40], and CB activation inhibits human sperm motility [41], primarily though CB1 [35, 38, 40, 42–45]. ECB-mediated effects on sperm motility may be explained from a metabolic standpoint considering that ECBs and phytocannabinoids alter mitochondrial $O_2$ consumption [46], membrane integrity [47], and electrochemical potential in human spermatozoa [34, 35, 48]. ECB-related changes to sperm mitochondrial membrane potential (MMP) are also correlated with viability, given that CB1 mediates pro-apoptotic MAPK signalling and ceramide production [32, 49–51]. ECBs influence several other essential processes for fertilization including capacitation, through CB1-mediated changes in cAMP levels [52], and inhibition of the AR by transient receptor potential vanilloid 1 (TRVP) activation [42, 53]. CB stimulation has also been implicated in modulating epigenetic factors in spermatozoa that influence placental and embryonic development [54–58]. Therefore, disruption of ECS signalling in sperm by THC may contribute to pathophysiological processes impacting sperm function [21, 39], acquisition of fertility [37], and embryo development [39, 42].

Intrinsic sperm factors, such as coding and non-coding RNAs, are transferred to the oocyte during fertilization and are an emerging field of interest for male fertility potential. Sperm-borne microRNAs (miRs) are important epigenetic factors transferred to the oocyte that function as post-transcriptional modulators of genes required for early embryonic development [6, 59–61]. Considering that 30% of human genes are regulated by miRs [62], aberrant expression may rapidly alter the proteome of cells, and lead to the inheritance of maladaptive paternal phenotypes [63–66]. Multiple groups have shown that miR profiles significantly differ between high and low fertility sperm from both humans and bovine [60, 61, 67–78], suggesting that these molecules may be used as biomarkers for fertility status. Specific miRs are known to be essential for embryo development. For example, miR-34c is the most abundantly expressed miR in human sperm and is essential for first cleavage in mice [60, 79], while sperm-borne miR-216b modulates KRAS—an essential protein for cell proliferation and differentiation in two-cell embryos [69]. Epigenomic components of spermatozoa, such as DNA methylation, are reactive to cannabis [54, 56, 80–82]; however, there is limited research investigating this relationship in terms of miR-profiles [22]. A transcriptomic analysis previously conducted by our group demonstrated that exposing sperm to THC significantly changed the abundance of several fertility-associated miRs (miRs-346, -324, and -33b) [22], suggesting that sperm miR-profiles may serve as molecular signatures indicative of specific morpho-functional abnormalities or contact with chemicals affecting reproduction.

Due to ethical constraints concerning the use of human reproductive tissue (regarding the in vitro production of embryos), the present study utilized a bovine model to investigate the effects of THC on sperm function and embryo development. Relative to other species, bovine is most similar to humans in terms of sperm morphology [83, 84], insemination location [85], epigenome [86], and transcriptome [22, 87] as well as oocyte size, maturation time, and early embryo development dynamics [88]. Human and bovine also share many of the same ECS characteristics within both sperm and the female reproductive tract [35, 89].

The present study aimed to further investigate the molecular and cellular effects of THC on morpho-functional and intrinsic features of sperm, and how these may impact embryo

development following IVF. To our knowledge, this is the first study to explore the impact of THC-exposed sperm on embryo development, and the abundance of miRs associated with fertility in both sperm and blastocysts. We examined sperm features including the AR, MMP, and miR profiles following THC exposure, as well as developmental rates, quality, and miRs in embryos derived from IVF with THC-treated sperm. We hypothesized that exposure of sperm to physiologically relevant concentrations of THC would negatively affect morpho-functional features, including the AR, MMP, and fertility-associated transcript levels, subsequently impacting embryo development, quality, and miR expression following IVF.

## Materials and methods

### Ethic statement

This research was carried out in accordance with the recommendations of the Animal Care Committee at the University of Guelph and adheres to the principles espoused by the Canadian Council on Animal Care (CCAC) [90]. This article does not contain any studies involving live animals; thus no further ethics approvals were required.

### Reagents

All chemicals and media were purchased from Sigma Aldrich (Oakville, ON, Canada) unless otherwise specified.

### Sperm preparation and treatment

Cryopreserved bull semen obtained from fertile bulls was thawed in 38.5˚C water. For each bull, 200µL of semen containing approximately 50 million sperm were thawed and washed using a discontinuous Percoll microgradient described by Truong et al. (2023) and then divided into five treatment groups: Control (HEPES/Sperm TALP supplemented with 0.3% bovine serum albumin), Vehicle (0.01% ethanol diluted in the Control media), High-THC (4.8µM), Mid-THC (0.32µM), and Low-THC (0.032µM). All THC concentrations are based on those used by Whan et al., (2006), reflecting mean plasma levels following therapeutic (0.032µM), and low (0.32µM) to high (4.8µM) recreational cannabis use in humans [22]. Sperm were incubated for 6 hours at 38.5˚C and 5% $CO_2$ [22].

### Assessment of sperm acrosomal reaction

Sperm acrosomal and plasma membrane (PM) integrities were assessed using co-staining with fluorescein isothiocyanate-conjugated-peanut agglutinin (FITC-PNA) (Millipore Sigma; L73811MG) and propidium iodide (PI) (Sigma Aldrich; P4170), respectively. FITC-PNA binds to specific glycoproteins on the outer acrosomal membrane, enabling comparison of reacted and unreacted sperm in a sample [91]. Simultaneous evaluation of acrosomal status and viability based on PM integrity was achieved by co-staining sperm with FITC-PNA and PI, a membrane-impermeable dye that binds to double-stranded DNA when the PM is compromised [92–94]. An additional positive control group was treated with 10µM of calcium ionophore (A23187) to induce the AR 15 minutes prior to analysis. After a 6-hour treatment period, 200µL samples containing 1–2 million sperm were co-stained for 15 minutes with a mixture of 1.5µg/mL of FITC-PNA and 1µg/mL PI at 38.5˚C and 5% CO2 in the dark. Before analysis, sperm were filtered through a 30µM cell strainer (Pluriselect; 43-50030-03) and kept at 38.5˚C in the dark. Flow cytometry using a C6 BD Accuri flow cytometer (BD Biosciences) allowed for the detection of the following sperm populations within each sample: 1) live, non-acrosome-reacted (FITC-PNA⁻/PI⁻); 2) live, acrosome-reacted (FITC-PNA⁺/PI⁻); 3) necrotic,

non-acrosome-reacted (FITC⁻PNA⁻/PI⁺); and 4) necrotic, acrosome-reacted (FITC-PNA⁺/PI⁺). FITC-PNA was detected using FL1 (533/30nm) and PI was detected on FL3 (>670nm). At least 25,000 events from each group were analyzed after gating using FlowJoTM v10 (BD Biosciences), and the flow cytometer was validated before each use.

## Assessment of sperm MMP

MMP was detected by staining sperm from each treatment group with 5,5′,6,6′-tetrachloro-1,1′,3,3′-tetraethylbenzimida-zolylcarbocyanine iodide (JC-1) (Fisher Scientific; 5016951)–a lipophilic, cationic dye that enters the mitochondria of healthy, motile sperm, and forms aggregates, which emit an orange fluorescence. When MMP is low, JC-1 remains in its mono-meric form, emitting a green fluorescence [95], meaning that the percent of sperm emitting orange fluorescence represents the proportion of sperm with high MMP [96, 97]. A positive control group was treated with 0.132mM of carbonyl cyanide 4-(trifluoromethoxy) phenylhy-drazone (FCCP) (Abcam; ab120081)–a potent mitochondrial oxidative phosphorylation uncoupler. After the 6-hour treatment period, 200μL samples containing 1–2 million sperm were stained with 1.7 μM of JC-1 for 15 mins at 38.5°C and 5% $CO_2$ in the dark. Before analy-sis, sperm were filtered through a 30μM cell strainer (Pluriselect; 43-50030-03) and kept at 38.5°C in the dark. Using a C6 BD Accuri flow cytometer (BD Biosciences), the percentage of sperm staining orange or green was quantified. At least 25,000 events from each group were analyzed after gating using FlowJoTM v10 (BD Biosciences), and the flow cytometer was vali-dated before each use.

## Mechanistic evaluation of THC-induced reduction in MMP with CB-antagonists

Given that THC can act as both a partial agonist and antagonist at CB-receptors [29], the pre-vious set of experiments measuring MMP were repeated with CB-receptor antagonists and only the High-THC treatment (shown to significantly reduce MMP). Sperm treatment groups included: 1) Control (HEPES/Sperm TALP + 0.3% BSA); 2) Vehicle (0.01% ethanol diluted in Control); 3) High-THC (4.8μM THC); 4) CB1 Antagonist (4.8μM SR141716 + HEPES/Sperm TALP + 0.3% BSA); 5) CB2 Antagonist (4.8μM SR144528 + HEPES/Sperm TALP + 0.3% BSA); 6) High-THC (4.8μM THC) + CB1 Antagonist (4.8μM SR141716); 7) High-THC (4.8μM THC) + CB2 Antagonist (4.8μM SR144528). SR141716 and SR144528 are selective CB1 and CB2 antagonists, respectively (Research Triangle Institute (RTI) International, NC, USA; 158681-13-1, 192703-06-3). Both antagonists were dissolved in the same media as THC prior to being added to their respective treatment groups.

## In vitro maturation of cumulus-oocyte-complexes

Bovine (*Bos taurus*) ovaries from a local abattoir (Cargill Meat Solutions, Guelph, ON, Can-ada) were aspirated using a vacuum apparatus and collected in a vacutainer tube with 1mL of 38.5°C oocyte collection media: 1M HEPES-buffered Ham's F-10 media (Sigma Aldrich; N6635) supplemented with 2% steer serum, heparin (2 IU/mL), sodium bicarbonate, and 1% penicillin/streptomycin (Gibco, Whitby, Canada; 15140–122). High-quality cumulus-oocyte-complexes (COCs) containing dark, homogenous oocytes surrounded by tightly packed cumulus cells were separated from debris and placed into in vitro maturation media (S-IVM), consisting of HEPES-buffered TCM199 maturation media supplemented with 2% steer serum and sodium pyruvate. COCs were washed and matured in 80μL micro-droplets of S-IVM+H, containing 10μL luteinizing hormone (LH) (1μg/mL—NIH), 12.6μL of follicle stimulating hormone (FSH) (0.5μg/mL—Follitropin V), 10μL Estradiol (1μg/mL—SIGMA E2785), and

800μL Fetal Bovine Serum (FBS) (10%—Gibco 12483–020) to 10 mL of S-IVM. COCs were matured under Lite Oil (Life Global; LGOL-100) in groups of 15–20 COCs/drop and matured for 22–24 hours at 38.5°C and 5% CO2.

### In vitro fertilization

Matured COCs were washed and fertilized with sperm from each group. COCs were washed with HEPES/Sperm TALP and with 0.3% BSA and BSA-supplemented with IVF Tyrode albumin lactate pyruvate (TALP). COCs were then placed into 80μL drops of IVF TALP + 15% BSA in groups of 15–20 COCs/drop and returned to the incubator until IVF. After swim-up separation, the upper layer was reconstituted in warmed HEPES/Sperm TALP with 0.3% BSA and IVF TALP + 15% BSA. Sperm motility was confirmed before fertilizing the COCs at a concentration of 1 million sperm cells/mL/drop. COCs + sperm were incubated for 18 hours at 38.5°C and 5% $CO_2$, with 40 COCs fertilized by each group of treated sperm per IVF run.

### In vitro culture

Eighteen hours after fertilization, presumptive zygotes (PZs) were stripped of any remaining cumulus cells by mechanical disruption and washed in HEPES/Sperm TALP with 0.3% BSA and SOF media supplemented with sodium pyruvate, essential amino acids (Sigma Aldrich; M5550), non-essential amino acids (Sigma Aldrich; M7145), gentamicin (Sigma Aldrich; G1272), 15% BSA diluted in SOF, and 2% FBS. PZs were then transferred into 30μL drops of SOF supplemented with the previously mentioned components (Life Global; LGOL-100) in groups of 20–30 COCs/drop.

### Cleavage and blastocyst rates

Cleavage rates were measured 48 hours after fertilization by calculating the number of cleaved embryos divided by the total number of presumptive zygotes (PZs). Blastocyst rates were measured on day 8 post-fertilization by dividing the number of blastocysts by the number of cleaved embryos calculated on day two post-fertilization. Both cleavage and blastocyst rates were assessed within an hour of the fertilization time.

### Differential staining and cell-counts in blastocysts

On day 8 post-fertilization, blastocysts generated from THC-treated sperm were incubated in RNAse (Fisher Scientific; EN0531) (100mg/mL in PBS) for 60 minutes at 38.5°C. Blastocysts were then stained with PI (50μg/mL in 0.1% Triton) for 30 seconds at room temperature and placed in Hoechst stain (10μg/mL in 4% paraformaldehyde) for 15 minutes at room temperature. Blastocysts were mounted onto slides using DakoCytomation Flourescent Mounting Media (DakoCytomation, Carpinteria, California, USA; S3023). Blastocysts were imaged with an Olympus FV120 confocal microscope at 40x magnification using FlowView software (V4.0b). Cell counts of the trophectoderm (TE) (red), and total cells (blue) were carried out both manually and automated counting (via ImageJ software V2.9.0). The number of cells within the inner cell mass (ICM) was calculated by subtracting the number of TE cells from the total number of cells. Manual counts on a sample size of 9–14 biological replicates (blastocysts) were used for statistical analysis.

### RNA extraction and reverse transcription

Using a Qiagen miRNeasy Micro RNA extraction kit (Qiagen, Toronto, ON, Canada; 217084), total RNA was extracted from either 400μL of frozen sperm from each treatment

group or pools of five blastocysts generated from THC-treated sperm following the manufacturer's protocol. Total RNA was assessed for quantity and quality using a NanoDrop 2000c spectrophotometer (Thermo Fisher Scientific; Whitby, ON, Canada) and then stored at -80˚C until reverse transcribed. Reverse transcription was performed using the miRCURY LNA Reverse Transcription (RT) Kit (Qiagen, Toronto, ON, Canada; 339340) and a T100 Thermal Cycler (BioRad; Mississauga, ON, Canada). 100ng of RNA from each sample was reverse transcribed by miRCURY SYBR® Green RT Reaction Buffer and miRCURY RT Enzyme Mix. The RT protocol consisted of 60 min at 42˚C, 5 min at 95˚C, and held at 4˚C till further analysis.

## Reference gene selection and quantitative polymerase chain reaction

qPCR was performed using a miRCURY LNA qPCR Kit (Qiagen, Toronto, ON, Canada; 339320) according to the manufactures protocol. A CFX96 Touch Real-Time PCR Detection System (BioRad) was then used to quantify miRs 324, 346, 34c, and 33b in sperm from each treatment group, and miRs 324, 346, and 34c in blastocysts generated by IVF using sperm from each treatment group. Each well contained 3μL of cDNA template diluted 1:60 for sperm and 1:10 for blastocysts. qPCR consisted of 2 min at 95˚C, 40 repetitions of 10 seconds at 95˚C followed by 60 seconds at 56˚C. Standard curves were used to determine the efficiencies of primers (Table 1). Relative changes in miR expression were calculated using an efficiency-corrected method (ΔΔCt) with miR-132 and miR-93 as reference targets. geNorm software and qPCR, adhering to MIQE guidelines [98], were used to select miR-132 and miR-93 as reference genes (S1 Fig). Control group RNA from either blastocysts or sperm was used as a calibrator to account for inter-run variability. A minimum of three biological replicates in technical triplicates was used to quantify each miR.

## Statistical analysis

GraphPad Prism 8 (Version 8.4.3) and SPSS statistics software (Version 28.0.1.1) were used to analyze difference amongst groups. All data were subjected to the Kolmogorov-Smirnov test for normality. Normally distributed data was analyzed using a one-way analysis of variance (ANOVA) followed by a Tukey's post-hoc test to determine differences between treatment groups. Non-normally distributed data sets were analyzed using a Kruskal-Wallis test. A minimum of three biological replicates was used and statistical significance was based on a two-tailed p-value <0.05. Data shown represents the mean ± standard error of the mean (SEM). No statistical differences were observed between control and vehicle groups throughout any experiment.

**Table 1. MicroRNA primers for qPCR.**

| MicroRNA | Primer ID | Accession # | Primer Sequence (5'-3') | Efficiency (%) |
|---|---|---|---|---|
| miR-324 | hsa-miR-324-5p | MIMAT0000761 | CGCAUCCCCUAGGGCAUUGGUG | 99.9% |
| miR-346 | hsa-miR-346 | MIMAT0000826 | UGUCUGCCCGCAUGCCUGCCUCU | 100.9% |
| miR-33b | hsa-miR-33b-5p | MIMAT0003301 | GUGCAUUGCUGUUGCAUUGC | 101.0% |
| miR-34c | hsa-miR-34c-3p | MIMAT0003247 | AAUCACUAACCACACGGCCAGG | 99.3% |
| miR-93 | hsa-miR-93-5p | MIMAT0000093 | CAAAGUGCUGUUCGUGCAGGUAG | 100.1% |
| miR-132 | hsa-miR-132-3p | MIMAT0000426 | UAACAGUCUACAGCCAUGGUCG | 100.3% |

* miR primers were predesigned and validated by Qiagen. Primer efficiencies were tested in the present study.

## Results

### Acrosomal status in THC-treated sperm

THC at any concentration did not significantly alter the percent of sperm belonging to live, non-acrosome reacted (FITC-PNA⁻/PI⁻; Fig 1A–1F and 1I); live, acrosome reacted (FITC-PNA⁺/ PI⁻; Fig 1G); necrotic, non-acrosome reacted (FITC-PNA⁻/PI⁺; Fig 1H), or necrotic, acrosome-reacted (FITC-PNA⁺/ PI⁺; Fig 1J) (n = 4). The positive control group, treated with 10μM of calcium ionophore (A23187), had significantly more acrosome-reacted-sperm compared to control and vehicle groups ($p < 0.05$; n = 3) (Fig 1F, 1G and 1I).

### THC reduces sperm MMP

Sperm treated with High-THC (4.8μM) had a significantly lower average percent of sperm with high MMP (37%) relative to control (52%) and vehicle (47%) groups ($p < 0.05$; n = 4) (Fig 2E and 2G). Sperm in the positive control group, treated with 0.132mM of FCCP, also had a significantly lower average percent of sperm with high MMP (18%) ($p < 0.05$) (Fig 2F and 2G). No significant differences in the percent of sperm with high MMP were observed for any other treatment group (Fig 2B, 2C and 2G).

### THC reduces sperm MMP via agonistic interactions

The previous set of experiments used to assess MMP were repeated with the High-THC group in addition to selective CB1 and CB2 antagonists, SR141716 and SR144528, respectively. As previously shown, the High-THC group had a significantly lower average percent of sperm with high MMP (38%) compared to control (51.5%) and vehicle (54.4%) ($p < 0.05$; n = 3) (Fig 3C and 3H). However, when treated with High-THC in combination with either CB receptor antagonist, there was no longer a significant difference in the percent of sperm with high MMP; High-THC + CB1 antagonist and High-THC + CB2 antagonist groups contained 47% and 48% of sperm with high MMP, respectively (Fig 3D, 3E and 3H). The greatest difference in the average percent of sperm with high MMP was observed between the CB1 antagonist (63.5%) and High-THC (38%) groups ($p < 0.00001$) (Fig 3C, 3F and 3H). The CB1 antagonist group also had a significantly greater average percent of sperm with high MMP (63.5%) compared to the High-THC + CB1 antagonist (47%), High-THC + CB2 antagonist (48%), and CB2 antagonist groups (46.6%) ($p < 0.01$) (Fig 3E–3H).

### IVF with THC-treated sperm does not affect developmental rates

Cleavage rates were calculated by dividing the number of 2-cell embryos by the total number of COCs fertilized 48 hours post-fertilization. On day 8 post-fertilization, blastocyst rates were calculated within 1 hour of the fertilization time as the number of blastocysts divided by the total number of cleaved embryos. A total of 240 COCs were analyzed per treatment group. There was no significant difference in either cleavage or blastocyst rates between any of the THC concentrations used to treat sperm (n = 3) (Fig 4A and 4B).

### IVF with THC-exposed sperm yields blastocysts with fewer cells

Blastocysts generated by IVF using sperm from each treatment group were collected on day 8 post-fertilization and stained with PI, allowing for peripheral staining of the TE (red), followed by Hoechst stain, which penetrated cells of both the TE (red) and ICM (blue). Between 9 and 14 blastocysts from each sperm treatment group were analyzed, with representative images depicted in Fig 5A–5E.

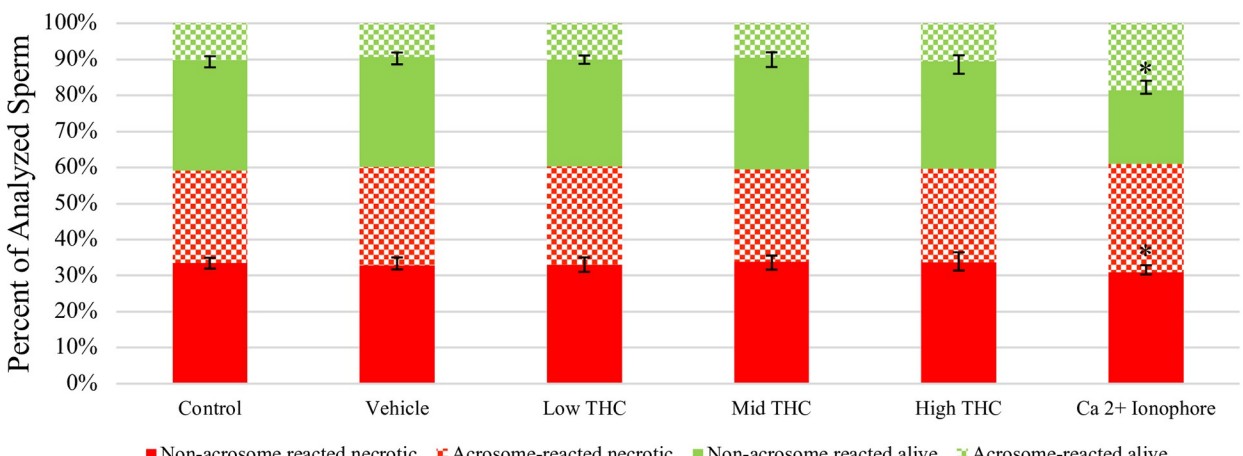

**Fig 1. THC-exposed sperm maintain acrosome integrity.** Representative scatter plots of sperm, co-stained with FITC-PNA and PI, and treated with (A) control, (B) vehicle, (C) Low-THC (0.032μM), (D) Mid-THC (0.32μM), (E) High-THC (4.8μM) or (F) calcium ionophore (10μM). The percent of sperm staining with FITC-PNA$^+$/PI$^-$, FITC-PNA$^+$/ PI$^+$, FITC-PNA$^-$/PI$^-$, or FITC-PNA$^-$/PI$^+$ (G), was measured using flow cytometry following a 6-hour treatment period. As a positive control, sperm were treated with a calcium ionophore for 15 mins prior to analysis. Bars represent ± SEM. * $p<0.05$.

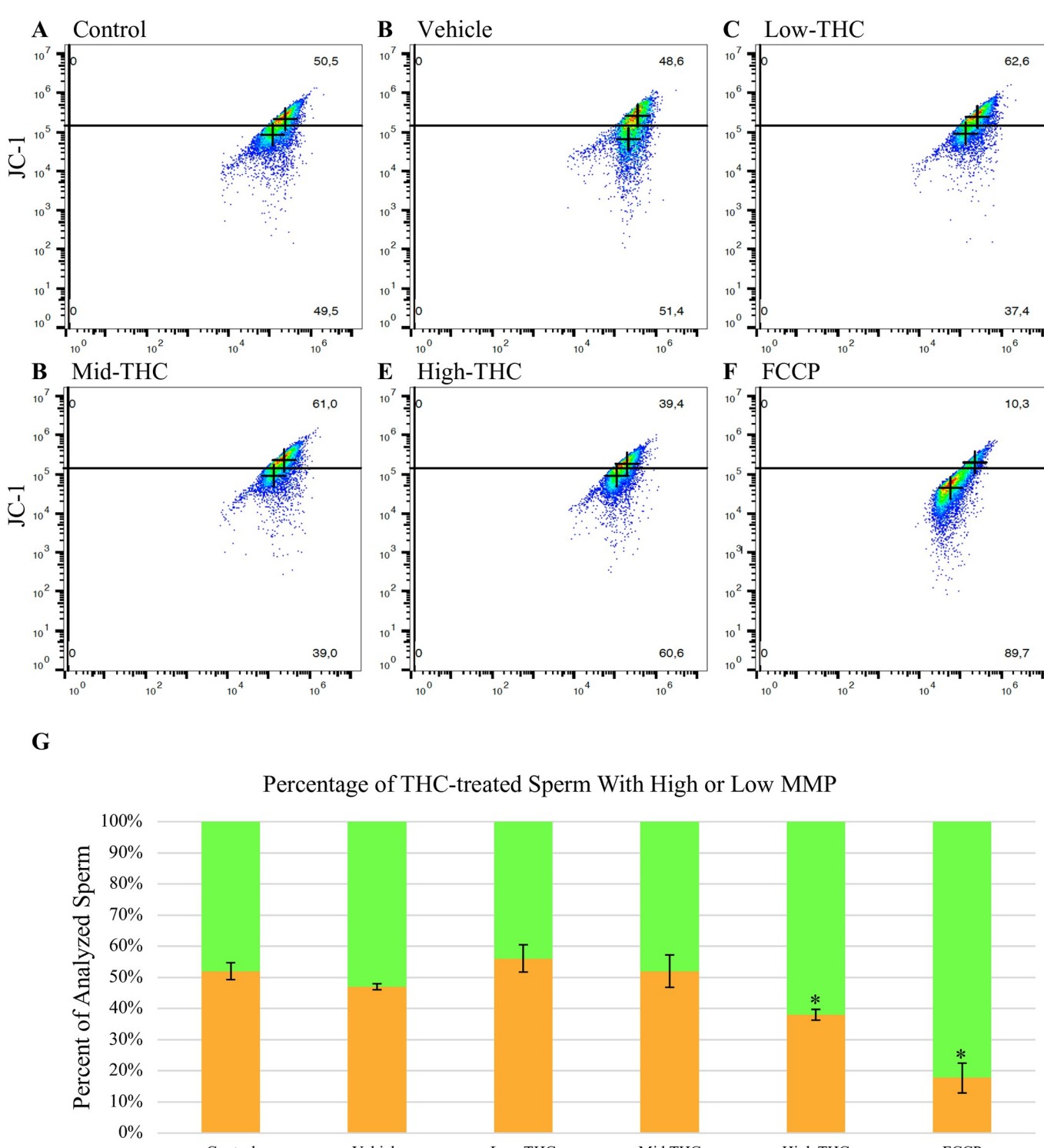

**Fig 2. THC-exposed sperm have lower MMP.** Representative scatter plots of JC-1-stained sperm with either high (top quadrant) or low (bottom quadrant) MMP treated with (A) control, (B) vehicle, (C) Low-THC (0.032μM), (D) Mid-THC (0.32μM), (E) High-THC (4.8μM) or (F) FCCP (0.132mM). (G) The percent of sperm with either high (orange) or low (green) MMP was measured using flow cytometry following a 6-hour treatment period. As a positive control, sperm were treated with FCCP for 15 mins prior to analysis. Bars represent ± SEM. * p<0.05.

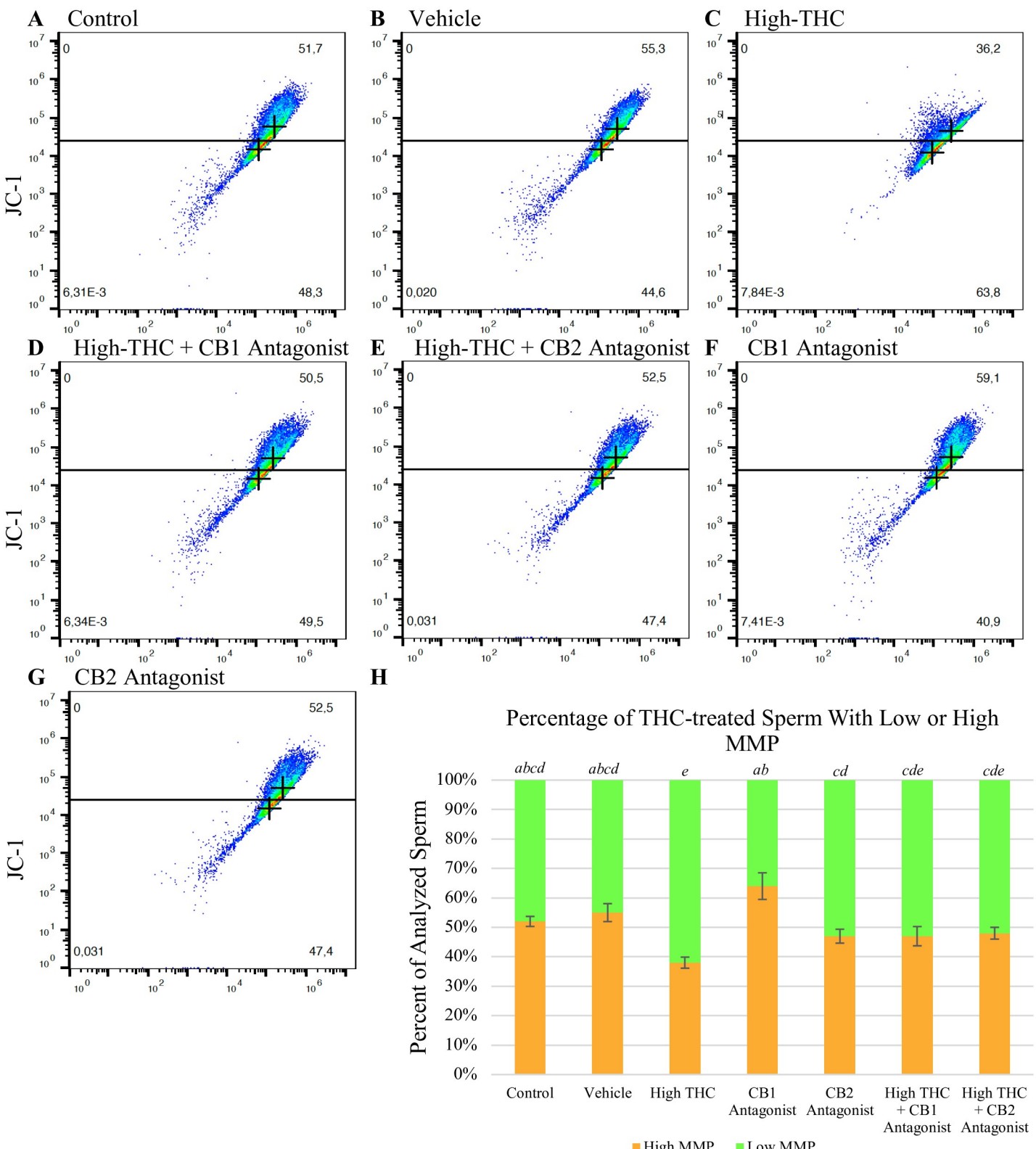

**Fig 3. THC reduces sperm MMP through agonistic interactions with CB-receptors.** Representative scatter plots of JC-1-stained sperm with either high (top quadrant or low (bottom quadrant) MMP treated with (A) control, (B) vehicle, (C) High-THC (4.8μM), (D) High-THC (4.8μM) + CB1 antagonist (4.8μM), € High-THC (4.8μM) + CB2 antagonist (4.8μM), (F) CB1 antagonist (4.8μM) or (G) CB2 antagonist (4.8μM). (H) The percent of sperm with either high (orange) or low (green) MMP was

measured using flow cytometry following a 6-hour treatment period. Bars represent ± SEM. Groups with different italicized letters contain significantly different percentages of sperm with high and low MMP.

After imagining blastocysts, cell counts were measured for the number of cells allocated to either the TE or ICM along with total cell counts (Fig 6). Blastocysts generated from sperm exposed to High-THC (Fig 5E) had a significantly fewer average total cells (103±8) compared to vehicle (145±11) (p<0.01) (Fig 6A). Similarly, blastocysts generated from sperm exposed to High-THC and Mid-THC (Fig 5D and 5E) had significantly fewer average TE cells (Mid-THC:78±7; High-THC:76±8) compared to the vehicle group (112±12) (p<0.05), but no difference was observed in blastocysts generated from sperm exposed to Low-THC (Figs 5C and 6B). Lastly, the average number of ICM cells in blastocysts generated from sperm exposed to Low (32±3; p<0.001), Mid (37±3; p<0.05) and High (34±4; p<0.05) THC were significantly lower than the control group (46±2) (Fig 6C). Interestingly, blastocysts generated from sperm exposed to Low-THC had the least number of ICM cells compared to blastocysts from the other treatment groups, resulting in the highest ratio of TE:ICM cells (Fig 6C and 6D).

## miR profiles of THC-exposed sperm

Levels of miRs-324, -346, -33b and -34c were quantified in THC-treated sperm using qPCR. Consistent with the transcriptomic analysis from Truong et al., (2023) [22], the abundance of miR-324, miR-346 and miR-33b were all significantly reduced in sperm treated with Mid-THC compared to vehicle (p<0.05) (Fig 7A, 7B and 7D). The abundance of miR-346 was also significantly reduced in sperm treated with Low-THC (p<0.05) (Fig 7B). Significantly higher levels of miR-34c were observed in sperm treated with Low-THC relative to vehicle (p<0.05)

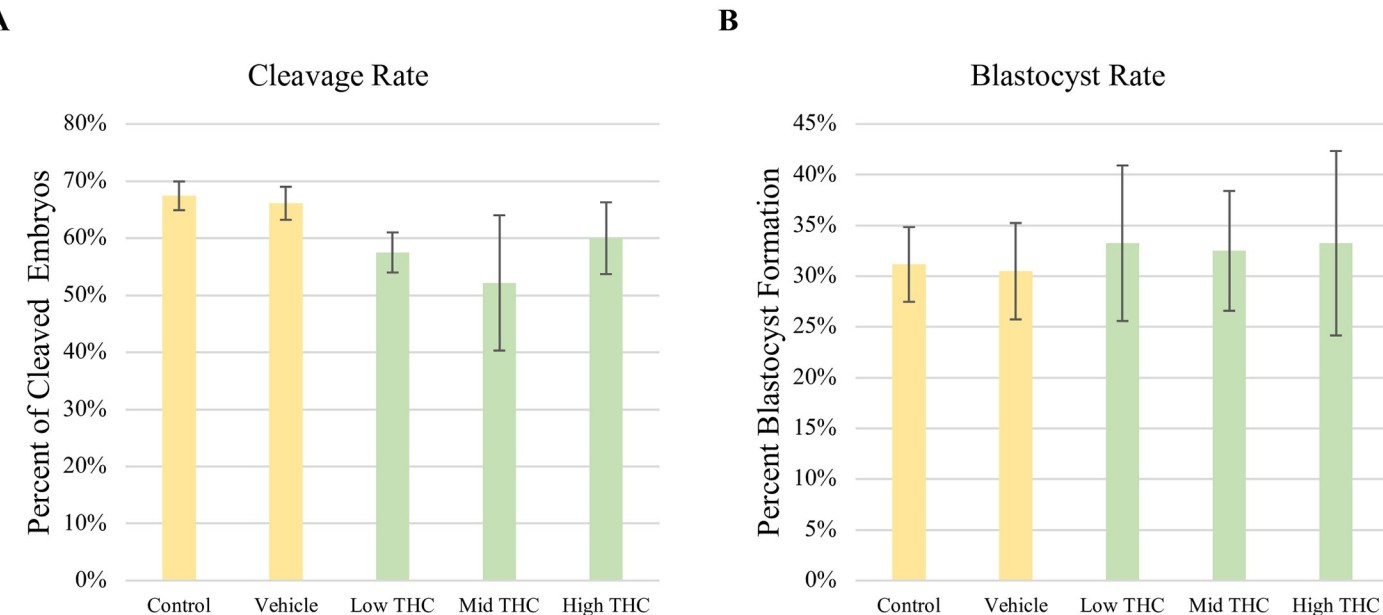

**A**

**B**

**Fig 4. Developmental rates of embryos following in vitro fertilization (IVF) with THC-treated sperm.** Sperm treated with control, vehicle, Low-THC (0.032μM), Mid-THC (0.32μM)), or High-THC (4.8μM) for 6 hours, were used to fertilize a minimum of 240 matured cumulus oocyte complexes (COCs) per treatment group. (A) Cleavage rates (number of 2-cell embryos / total fertilized COCs) were measured 48 hours post-IVF. (B) Blastocyst rates (number of blastocysts / total number of 2-cell embryos) were measured 8 days post-IVF. Bars represent ± SEM.

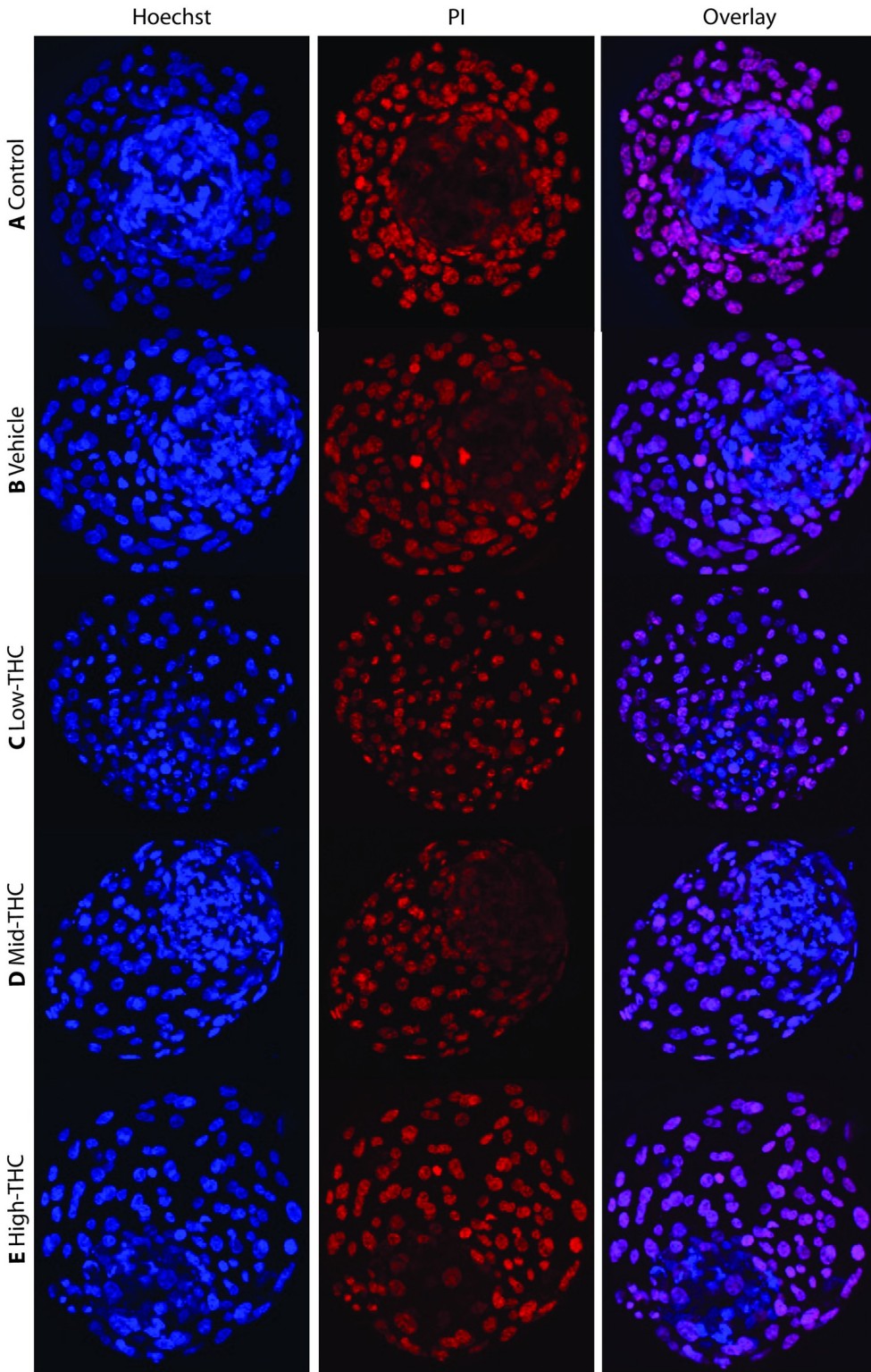

**Fig 5. Differentially stained blastocysts derived from THC-treated sperm.** Representative images of day-8 blastocysts derived from sperm treated with (A) control, (B) vehicle, (C) Low-THC (0.032µM), (D) Mid-THC (0.32µM), or (E) High-THC (4.8µM), for 6 hours prior to IVF. Hoechst stain (blue) penetrated all cells, while PI (red) strictly stained the trophectoderm (TE). Overlaying stains (pink) allowed for counting of cells strictly belonging to the inner cell mass (ICM) (blue). 9–14 blastocysts per treatment group were imaged using an Olympus FV120 confocal microscope at 40x magnification and analyzed using ImageJ software.

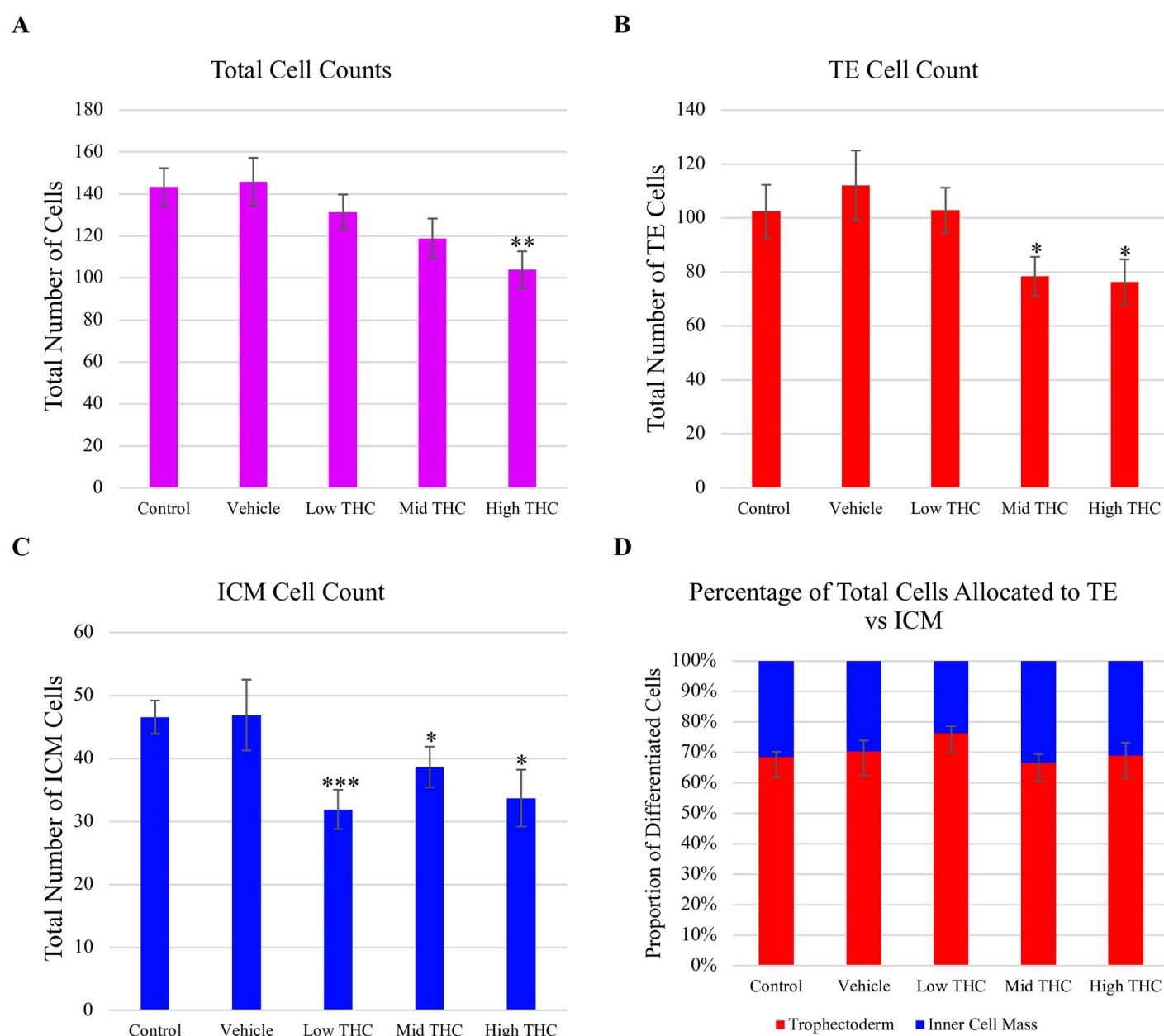

**Fig 6. Cell counts of the trophoectoderm (TE) and inner cell mass (ICM) in blastocysts derived from THC-treated sperm.** (A) Average total cell counts of day-8 blastocysts generated from sperm treated with control, vehicle, Low-THC (0.032μM), Mid-THC (0.32μM), or High-THC (4.8μM) for 6 hours prior to IVF. (A) Average total cell counts included a sum of the (B) average TE and (C) ICM cells per group. (D) The proportion of cells allocated to either the TE or ICM was also recorded. 9–14 blastocysts per treatment group were imaged using an Olympus FV120 confocal microscope at 40x magnification and analyzed using ImageJ software. Bars represent ± SEM. * p<0.05, ** p<0.01, *** p<0.001.

(Fig 7C). No significant differences in the abundance of miRs-324, -346, -33b and -34c were observed in sperm treated with any other concentrations of THC.

## miR profiles of blastocysts generated from THC-exposed sperm

Blastocysts were frozen for RNA extraction 8 days following IVF using THC-treated sperm. RNA from pools of 5 blastocysts was reverse transcribed and levels of miRs-324, -346, and

**A**

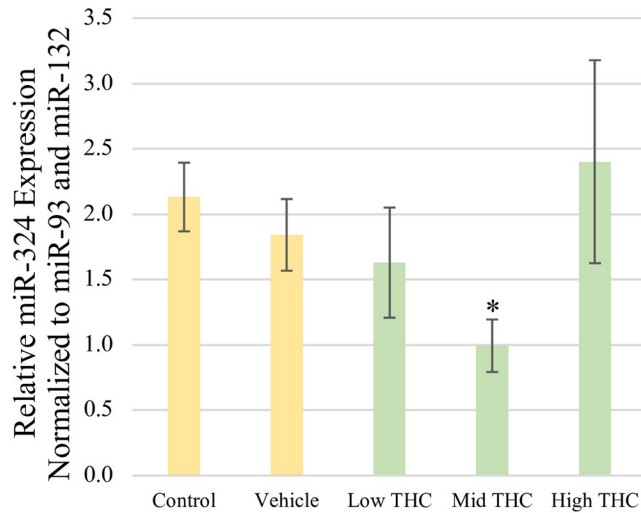

**B**

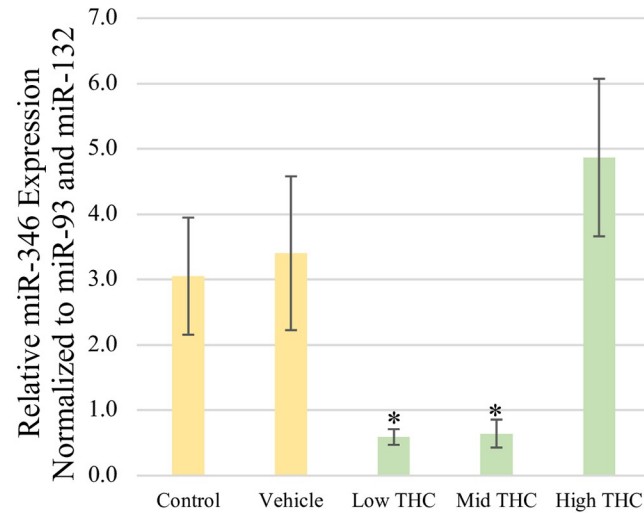

**C**

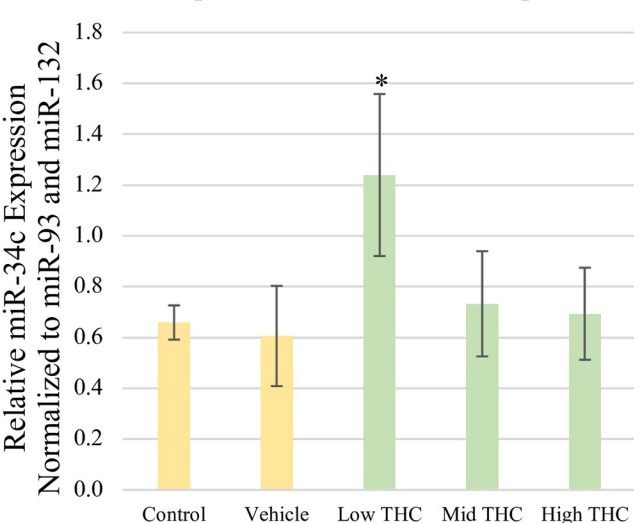

**D**

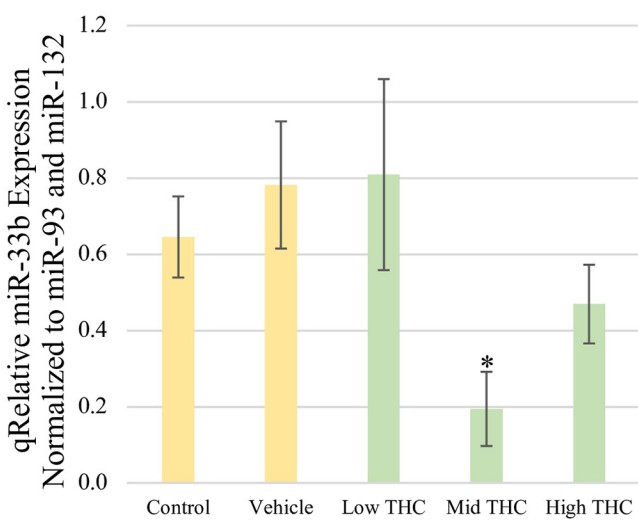

**Fig 7. Relative expression of fertility-associated miRs in THC-treated sperm.** Using qPCR, relative levels of (A) miR-324, (B) miR-346, (C) miR-34c, and (D) miR-33b were quantified in total RNA extracted from sperm treated with control, vehicle, Low-THC (0.032μM), Mid-THC (0.32μM), or High-THC (4.8μM) for 6 hours. The expression of all miRs was normalized to miR-93 and miR-132 as reference genes. Bars represent ± SEM. * $p < 0.05$.

-34c were quantified using qPCR (Fig 8). Compared to vehicle, levels of miR-346 were significantly increased in blastocysts generated from sperm exposed to Mid-THC ($p < 0.05$) (Fig 8B). However, no significant changes in levels of miR-346, miR-324 or miR-34c were observed in blastocysts generated from sperm exposed to any other concentration of THC investigated (Fig 8A and 8C). miR-33b was not detected in blastocysts.

**A**

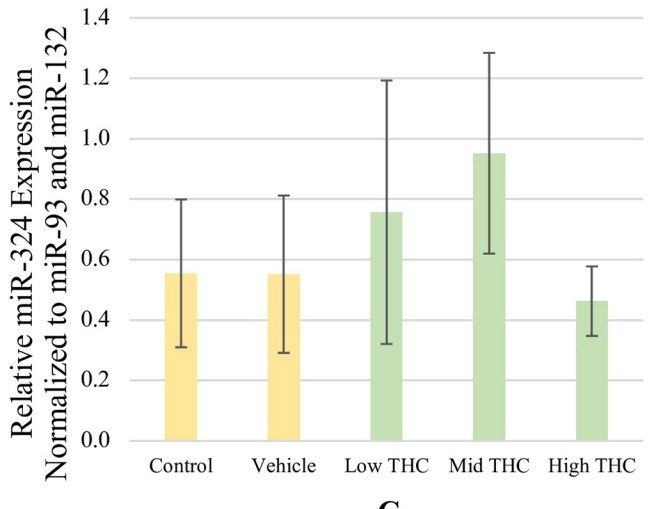

**B**

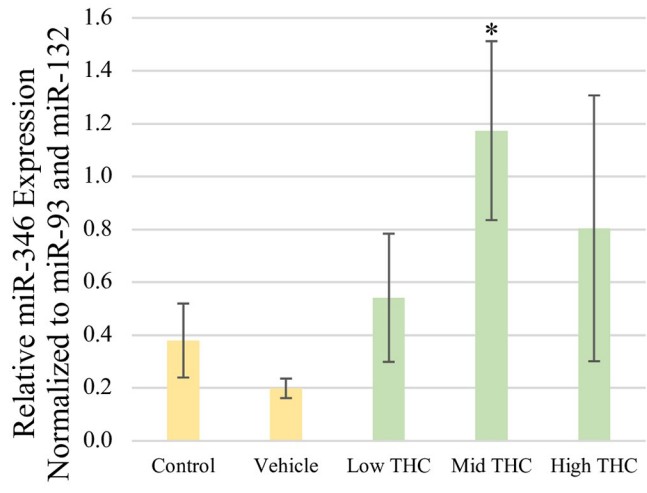

**C**

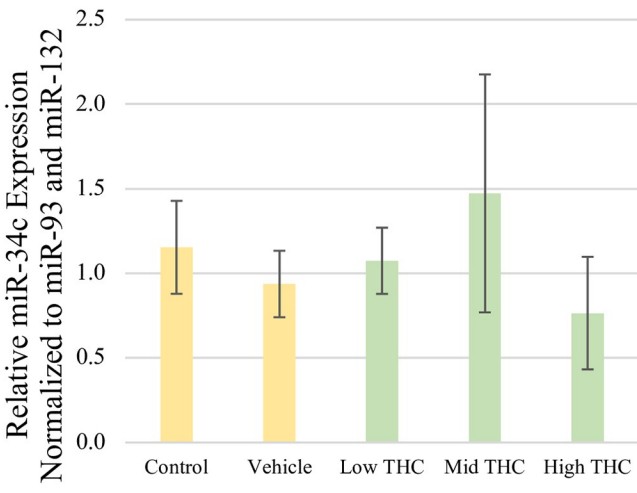

**Fig 8. Relative expression of fertility-associated miRs in blastocysts generated from THC-treated sperm.** Using qPCR, relative levels of (A) miR-324, (B) miR-346, and (C) miR-34c were quantified in total RNA extracted from 8-day blastocysts derived from sperm treated with control, vehicle, Low-THC (0.032μM), Mid-THC (0.32μM), or High-THC (4.8μM) for 6 hours prior IVF. The expression of all miRs was normalized to miR-93 and miR-132 as reference genes. Bars represent ± SEM. * $p < 0.05$.

## Discussion

This study investigated the AR, MMP, and fertility-associated miRs in THC-exposed sperm along with development rates, quality, and miRs in IVF embryos produced with THC-treated sperm (cryopreserved and thawed). In contrast to previous literature [21, 35, 42, 43], THC did not alter the percent of acrosome-reacted sperm. High-THC significantly reduced the average percent of sperm with high MMP, which is indicative of both human sperm motility and

viability [50, 51, 95, 99]. CB-antagonists (4.8uM) nullified the High-THC-induced decrease in MMP, suggesting agonistic interactions with CB-receptors. The largest difference in the percent of sperm with high MMP was observed between the High-THC and the CB1 antagonist groups. Fertilization of oocytes with THC-treated sperm did not alter cleavage or blastocysts rates, but blastocysts generated from sperm exposed to High-THC had significantly fewer total cells, trophoblasts and ICM cells. Blastocysts from the Mid- and Low-THC groups had significantly less ICM cells, while those from the Mid-THC group also had fewer trophoblasts. Consistent with prior research from our lab [22], miRs-324, -346 and -33b were significantly reduced in the Mid-THC-treated sperm. However, Low-THC significantly increased miR-34c and reduced miR-346. In contrast, blastocysts from Mid-THC-treated sperm had significantly higher levels of miR-346. Our data shows that THC lowers sperm MMP and changes fertility-associated miRs levels, possibly impacting embryo quality and miR-profiles following IVF, which could adversely impact implantation and pregnancy success.

THC doses used in the present study reflect plasma concentrations following therapeutic (0.032μM) and low (0.32μM) to high (4.8μM) recreational cannabis use [20, 100], as per Whan et al., (2006). Plasma concentrations of THC are moderately correlated with seminal levels, ranging from 0.87–0.97ng/mL in the semen of chronic cannabis users 10–12 hours after cannabis exposure [101].

Capacitation and the AR are heavily influenced by ECB signalling [102]. We found no impact of THC on the average percent of acrosome-reacted sperm, which is a reliable predictor of IVF capacity [103]. In contrast to our findings, others have demonstrated that in vitro exposure to 1.5μM, 0.032μM, and 4.8μM THC significantly reduced the percent of human acrosome-reacted sperm [21, 43], while 1.0μM of AEA has been shown to inhibit the AR [35].

On the other hand, 2-AG exposure increases the percent of human acrosome-reacted sperm [38]. Troung et al., (2023) noted that treatment of bovine sperm with 0.32μM THC significantly increased capacitation, which would simultaneously promote the AR [22]. Similarly, Gervasi et al., (2011) found that treatment of bull sperm with concentrations of AEA found in the female reproductive tract significantly increased the percent of capacitated and acrosome-reacted sperm by bolstering calcium influx and tyrosine phosphorylation [89]. AEA at nanomolar concentrations can release bovine sperm from oviductal epithelium without impacting the AR, suggesting that AEA-induced changes to the AR are dose-dependent [104]. Although THC does not interact with TRPV1, it likely shifts levels of bioavailable AEA by competitively binding CB1 receptors in sperm, which would significantly impact sperm function [27, 36, 92, 105].

THC and AEA have similar affinities for CB1; however, as a partial agonist, THC may not have stimulated CB1 enough to affect the AR in the present study. THC-induced effects on the AR are most pronounced in sperm of low quality [21]. The use of bovine sperm of proven fertility, having undergone multiple separations to select high quality sperm, could explain the lack of observed effects. Clinical data supports that the greatest risks of cannabis-related alterations to sperm function are among patients with borderline fertility issues [17]. The level of sperm processing used in the present study was similar to that conducted in fertility clinics.

Sperm mitochondria facilitate progressive motility by producing ATP used for flagellar beating [6, 50, 93, 105]. Both mitochondrial morphology and function are pre-requisites for fertilization across multiple species [6, 51, 95, 99, 106–110]. MMP typically ranges from 80-100mV in healthy sperm [51, 95], is regulated by ECBs [37, 46], and indicative of both human sperm motility and viability [50, 51, 95, 99]. Our results showed that a high recreational dose of THC (4.8μM) significantly reduced the percent of sperm with high MMP. Given that THC can also act antagonistically at certain concentrations [29], we repeated the same set of experiments with CB-antagonists to determine the pharmacodynamics underlying THC-induced MMP reduction. A THC concentration of 4.8μM in combination with 4.8μM of either CB-

antagonist eliminated the reduction in MMP, suggesting that THC may disrupt the electro-chemical gradient in sperm mitochondria though agonistic interactions with CB-receptors. The percent of sperm with high MMP differed the most between the high-THC and the CB1 antagonist groups, implying that THC may predominantly act at this receptor. This is consistent with research showing that THC has a greater affinity for CB1 than CB2, and interacts with mitochondrial CB1 (mtCB1) in sperm [26, 27, 105, 111, 112]. Interactions between spermatic mtCB1 and cannabinoids disrupt mitochondrial activity [34, 35, 44, 46, 48, 112]. In support of our findings, THC has been shown to impair mitochondrial respiration in human sperm [46] and lower MMP in cells from the placenta [113, 114], lung [115], and heart [116]. However, SR141716 –the CB1 antagonist used—can also act as an inverse agonist at CB1, meaning that changes to sperm MMP could also have been a result of supressing ligand-free signalling [117, 118]. Sperm are highly vulnerable to oxidative damage because they contain low levels of antioxidative enzymes and have high levels of lipids in their PM [119, 120]. In this regard, low sperm MMP can result from lipid peroxidation, which triggers apoptosis and a loss of progressive motility [120]. ROS-related damage also accounts for approximately 30–80% of male infertility cases [121, 122].

Low sperm MMP and subsequent loss of motility are primarily a result of decreased ATP availability. THC-induced decrease in mitochondrial ATP production occurs in other cell types [123], likely via CB1-mediated reduction in citrate synthase activity [124], which increases following exposure to CB1-antagonist, SR141716 [125]. Citrate synthase activity is a biomarker of mitochondrial membrane integrity, regulates the Krebs Cycle, and reductions to its activity may lower the $NAD^+$/NADH ratio in sperm.

Although CB2 is not located on sperm mitochondria, THC combined with a CB2 antagonist nullified the high-THC-induced reduction in MMP. In contrast to our results, Xu et al., (2016) showed that cell-impermeable CB-antagonists minimally affected mitochondrial respiration [126], suggesting that THC-mediated changes to MMP may result solely through cellular uptake and subsequent activation of mtCB1. However, CB2-induced reduction to human sperm motility occur in a manner distinct from CB1 [41]. In support of our findings, Herrera et al., (2006) reported that CB2 antagonism abolished pro-apoptotic effects of THC, including hypopolarization of the mitochondrial membrane and cytochrome-c release, in human leukemia cells that only express CB2 [127]. A THC concentration of 1.5µM, which is comparable to the high recreational dose used in the present study, reduced MMP and increased ceramide biosynthesis. Pan-caspase inhibition and selective caspase-8 inhibitors were unable to prevent THC-induced reductions in MMP, suggesting that CB2 receptor activation triggers apoptosis through a ceramide-dependent intrinsic mitochondrial pathway [127]. Treatment of mice sperm with and without CB1 with THC reduced ATP levels and progressive motility at concentrations as low as 1µM [128], suggesting that interactions between THC and sperm mitochondria are not CB1-dependent. Although mice sperm do not express CB2 receptors, sperm contain all the enzymatic machinery required for synthesizing ECBs, meaning that shifting one component of the ECS will impact homeostasis of the whole system. The "on-demand" nature of ECB production could mean that THC shifts the availability of AEA and 2-AG, contributing to the observed changes in MMP. In the context of humans and bovine, THC may lower sperm MMP through different mechanisms depending on whether it interacts with CB1 or CB2. Ultimately, THC-induced reduction in MMP may reflect multiple downstream events that compromise sperm progressive motility, viability, and fertilization potential. Given that sperm MMP is highly predictive of 4-hour progressive motility [95], some research suggests that measuring MMP is the most sensitive test to predict sperm quality and IVF success [129]. Based on our results, paternal THC-exposure could compromise the success of ARTs by impairing sperm mitochondrial function.

Adequate morpho-functional and intrinsic sperm features are predictive of successful in vitro embryo production [130–133]. Fertilizing oocytes with defective sperm negatively impacts embryo development [134], while micro-injecting oocytes with morphologically damaged sperm impairs mitotic division [130]. Low sperm quality reduces blastocyst formation rates [131, 132], whereas specific tail and head defects, as well as a high Multiple Abnormalities Index (MAI), are associated with poor embryo morphokinetics and implantation success [133]. We found that fertilization of oocytes with THC-treated sperm did not alter developmental rates, but blastocysts generated from sperm exposed to THC had significantly fewer total cells, trophoblasts and ICM cells, suggesting that THC-exposed sperm may impair proper cell division and the formation of high-quality blastocysts.

The evaluation of embryo quality prior to transfer is important for successful implantation and pregnancy, and for improving embryo culture systems [135–137]. First proposed by Rehman et al., (2007), blastocyst quality score (BQS) numerically classifies blastocysts based on morphology parameters described by Gardner et al. [135, 138–141]. Studies show a strong correlation between BQS and differential cell counts of the TE and ICM in blastocysts [135, 142]. According to Thompson et al., (2013), embryo score is significantly correlated with live birth outcomes, with 50% of hatched blastocysts, 49.5% of expanded blastocysts, and only 36.7% of early blastocysts (poor quality) resulting in live births [143]. Cell counts are also indicative of bovine embryo quality and future pregnancy outcomes [144–148]. A higher number of tightly packed ICM cells is significantly correlated with a higher embryo score and more likely to result in a successful pregnancy [146]. On the other hand, a high-quality TE, containing a cohesive epithelium formed by many cells, is indicative of live birth outcomes with 50% of good TE embryos, 41.9% of fair TE embryos, and 30% of poor TE embryos resulting in live births [143].

Given that blastocysts generated from sperm exposed to all concentrations of THC had fewer ICM cells, and that ICM quality is highly predictive of pregnancy and implantation rates [137], paternal cannabis use would be expected to adversely affect IVF outcomes. However, current clinical research shows no associations between paternal cannabis use and implantation rates or time to pregnancy [24, 25]. In support of our results regarding cleavage and blastocyst rates, Har-Gil et al., (2021) observed no differences in blastocyst formation rates between cannabis users and non-users seeking infertility treatment [25]. In contrast to our findings, the same researchers also found no effect of paternal cannabis use on blastocyst quality. Since the legalization of cannabis in various countries, Har-Gil et al., (2021) has been the only study to investigate cannabis use and ART success, emphasizing the need for more recent research that considers current THC levels in cannabis [25].

Composite measures incorporating the amount and timing of paternal cannabis use indicated an 11% reduction in birth weight among babies from male cannabis users compared to non-users when controlling for cigarette use, age, and socioeconomic status [149]. Considering that ICM grade/quality is predictive of birth weight [136], associations between male cannabis use and reduced birth weight [149] could be explained by our results, which show that blastocysts generated from sperm exposed to therapeutic and recreational doses of THC had significantly fewer ICM cells.

To our knowledge, the present study is the first to measure changes in early embryo development following fertilization with sperm exposed to THC in vitro. Some in vivo studies in mice investigated the effects of paternal cannabis exposure on embryo development. Morgan et al., (2012) found that acute injections of 50mg/kg of THC into male mice caused a 20% decline in the number of day-12 embryos [128]. As this effect did not occur in CB1 knockout males, THC-induced changes to male fertility may be CB1-dependent. Acute administration of THC also means THC-induced changes to embryo development were likely independent of

endocrine-related mechanisms. In support of Morgan et al., (2012), others report that paternal administration of 50mg/kg of THC caused defects in 33% of litters [150]. On the other hand, López-Cardona et al., (2018) found that chronic injections of male mice with 10mg/kg of THC did not significantly affect sperm motility, fertilization, or embryo production [151]. Given that THC has different effects on reproduction when administered acutely or chronically [152], the discrepancy between these results may be due to different administration protocols. Using an in vitro model, our results agree with components of both studies [128, 151]. However, neither López-Cardona et al., (2018) or Morgan et al., (2012) investigated embryo quality following IVF with sperm from THC-treated mice. Additionally, THC concentrations in mice models are typically quantified in mg/kg of body weight, while standard dosing in humans is not body-mass specific, with upper therapeutic thresholds ranging between 20–40 mg/day [153].

In somatic cells, CB activation alters the expression of coding and non-coding genes [154–156]. Although sperm are mainly transcriptionally silent, a transcriptomic analysis previously conducted by our group demonstrated that exposing sperm to THC significantly changed the abundance of several fertility-associated miRs, including miR-346, miR-324, and miR-33b [22]. Mature, sperm-borne miRs are transferred to the zygote and dictate the success of embryonic development [68, 73, 74, 76, 77, 157–161]. Few have simultaneously evaluated cannabinoid-induced changes to fertility-associated miRs in sperm and embryos. While our results agree with the findings of Truong et al., (2023) [22], we also observed that miR-346 was significantly reduced in sperm treated with Low-THC. In contrast, miR-34c abundance increased in sperm treated with low THC. Salas-Huetos et al., (2016) observed that sperm from normozoospermic infertile individuals had greater levels of miR-324 and lower levels of miR-346 [68], while Alves et al., (2019) found that mir-33b levels were higher in high fertility bovine sperm [69]. Truong et al., (2023) observed that sperm treated with 0.32μM THC had reduced levels of miRs-324, -346, and -33b, and were more likely to undergo premature capacitation [22]. In other cell types, miR-324 has been shown to be protective against hypoxia/reoxygenation-induced damage by regulating NF-kB/TNF-alpha signaling, where increased miR-324 expression reduced TNF-alpha and apoptosis [162, 163]. Considering that TNF-alpha mRNA is present at greater levels in the seminal plasma of infertile men [164] and interferes with embryo development [165], THC-induced reductions in sperm miR-324 may influence the abundance of this cytokine in sperm and embryos.

Associations between sperm miR-34c abundance, sperm quality, and ART outcomes are not consistent across species. Liu et al., (2012) showed that higher levels of miR-34c were associated with higher fertility in mouse, while Fagerlind et al., (2015) reported that lower levels were associated with high fertility in bovine [60, 74]. In humans, miR-34c is significantly lower in sperm from men diagnosed with different forms of spermatogenic abnormalities, including oligozoospermia, asthenozoospermia, teratozoospermia, oligoasthenoteratozoospermia, idiopathic male infertility, and in the seminal plasma of obstructive and nonobstructive azoospermia [73, 166–169]. Levels of miR-34c in sperm have also been correlated with ART outcomes. Cui et al., (2015) observed that sperm miR-34c levels were positively correlated with the number of high-quality human embryos implantation, pregnancy, and live birth rates [61]. Similarly, Shi et al., (2020) observed a negative correlation between spermatogenic miR-34c levels and embryo developmental kinetics, but a positive correlation with blastocyst formation rates, number of high-quality blastocysts, and pregnancy [170]. Human sperm containing higher levels of miR-34c had a 14-fold increased probability of obtaining a viable embryo [73]. Interestingly, miR-34b/c levels in sperm were only associated with a decrease in miscarriages, and increased implantation and pregnancy rates at certain Ct thresholds, determined using receiver operating characteristic curve (ROC) analysis [171]. Mechanistically, miR-34c-related

improvements to pre-implantation embryo development appear to be stage-dependent and occur through diverse biological processes [172].

Our results suggest that in vitro exposure to THC alters fertility-associated miR-profiles in sperm both favorably and unfavorably depending on the concentration. Determining the molecular mechanisms involved in correlative observations between miR-profiles in sperm and fertility would require functional studies designed to evaluate the individual importance of miRs-346, -324, -33b and -34c, along with their downstream targets. However, miRs work in association with each other, within complex regulatory networks of genes, where one miR may have multiple mRNA targets, making it difficult to determine their individual importance. For example, the activity of miR-34c and impact on embryo development is highly related to the expression of miR-449b [157]. The clinical relevance of spermatic miR abundance to fertility outcomes appears to be dependent upon certain thresholds, which may change based on the miR under investigation [171]. Despite positive correlations between elevated spermatic miR-34c and embryo quality, it is possible that we did not observe any improvement in the quality of blastocysts generated from sperm with Low-THC-induced increases in miR-34c levels because this change did not reach a clinically relevant threshold [171]. Nonetheless, the fact that THC alerted the abundance of fertility-associated miRs in sperm provides evidence that THC can influence important epigenetic modulators that are passed on to the developing embryo, some of which may become predictive biomarkers of defective sperm or exposure to reproductive toxicants.

The THC-induced reduction in miR-346 levels in sperm might have caused a lower amount of this miR to be transferred to the oocyte during fertilization, requiring transcriptional compensation from the blastocyst, thus explaining the detected increase in blastocysts. The low-density lipoprotein receptor-related protein 6 (LRP6) gene is a target of miR-346 and responsible for trophoblast proliferation and migration [173]. Zhang et al., (2020) show that upregulation of miR-346 is accompanied by lower LPR6 expression—an effect that is reflected in trophoblast activity. On a molecular level, these results provide one possible explanation for our observation that blastocysts produced from sperm exposed to a mid-recreational dose of THC had fewer trophoblast cells and higher levels of miR-346 [173].

miR-346 also directly regulates androgen receptor (AR) expression in prostate cancer cells, where inhibition of miR-346 significantly increases AR transcriptional activity, mRNA, and protein levels [174]. ARs are present within the ICM of blastocysts and during pre-implantation embryo development [175]. Stimulation of ARs may lead to apoptosis, while antiandrogens promote cell growth [175] and have been shown to reduce oocyte and embryonic degeneration [176]. The upregulation of miR-346 in blastocysts generated from sperm exposed to a Mid-THC treatment could explain why these blastocysts had fewer ICM cells. Considering that AR expression is critical for Sertoli cell maturation, blood-testis-barrier (BTB) formation and maintenance, germ cell proliferation and differentiation, THC-induced reduction in miR-346 transcripts may also explain the association between cannabis use and lower sperm counts [177]. Considering that the contribution and abundance of sperm-borne miRs during pre-implantation embryo development are stage-dependent [172], it is possible that we found no significant differences in the abundance of miRs-324, and -34c in blastocysts generated from THC-treated sperm because we quantified miRs only at the blastocyst stage. Although we analyzed RNA from whole blastocysts, differentially expressed genes predictive of pregnancy success appear to be more abundant within the TE compared to the ICM in human blastocysts [178].

Despite sperm being transcriptionally inactive, their mitochondria contain ribosomes capable of translation of both mitochondrial and nuclear transcripts, the latter being less well understood [179]. Inhibition of mitochondrial translation results in decreased capacitation,

motility, and fertilization potential [179]. Mitochondrial translation and subsequent transcript turnover may, in-part, explain the mechanism responsible for the THC-induced changes in miR abundance we observed. Truong et al., (2023) found that exposure of bovine sperm to Mid-THC changed the abundance of several genes including sperm acrosome-associated protein 7 (SPACA7) and FAD-linked sulfhydryl oxidase ALP (GFER), both of which are actively transcribed by mitochondrial ribosomes [22, 179]. Changes in mitochondrial ribosomal function would also be expected to result from THC-induced changes to mitochondrial activity, described above [180]. Transcript degradation is a possible alternative mechanism responsible for changes to the abundance of miR between THC treatments. Although transcript degradation in sperm has been previously described in mammals, it remains poorly understood [181].

The use of bovine sperm and oocytes as a translational model for humans could be considered a limitation of the present study due to potential species differences in responses to THC. The sperm used in this study were also cryopreserved to commercial standards, eliminating non-viable/substandard sperm, round somatic cells, debris, and seminal plasma—components that would normally exist in semen of cannabis users, making our results even more conservative as they would probably be further significant if experiments were performed in human sperm with a higher degree of individual variability in quality and fertility potential. On the other hand, cryopreservation may have damaged the sperm, making them either more or less susceptible to THC-induced changes. Cryopreservation is routinely employed in both human and bovine ART practices for reasons, such as fertility preservation, despite the possible damaging effects it can have on sperm (e.g. loss in motility and plasma membrane functionality) [182]. Furthermore, cryopreserved sperm is often used in research to study the effects of various drugs and chemicals on sperm function due to the availability and ease of storing a high number of samples. Embryo quality is based on a variety of morphological and genetic features, meaning that investigating additional parameters such as apoptosis, cellular fragmentation, and aneuploidy rates may provide a more comprehensive view of how THC-treated sperm impact embryo development.

With accelerated declines in global sperm quality, biomonitoring studies should be integrated with epidemiological data to determine how changes in sperm quality may be related to environmental and lifestyle factors such as cannabis use. Our data support that THC at a high recreational dose adversely affects sperm mitochondria, and that fertilizing oocytes with sperm exposed to therapeutic and mid-high recreational doses of THC reduces embryo quality following IVF. Our results also indicate that THC-induced alterations to sperm miR profiles may either promote or hinder fertility, associated with changes in levels of miRs-346, -324, 33b, and 34c. Only miR-346 expression was changed in blastocysts generated from THC-treated sperm, whereas THC did not alter the acrosomal reaction in sperm, contradicting prior literature [21]. The most clinically relevant outcomes of our study pertain to reduced embryo quality following IVF with THC-exposed sperm, suggesting that paternal cannabis use may compromise the success of ARTs.

## Supporting information

**S1 Fig. Optimal reference gene selection for THC-treated sperm and associated blastocysts following IVF.** (A) Average stability of 8 candidate genes: U6, miRNA-320a, miRNA-103a, Let7a, miRNA191, miRNA-106a, miRNA-132, and miRNA-93. (B) Determination of the optimal number of reference ttargets with a V vale set at 0.15, showing that 2/3-7/8 reference genes can be used to obtain the approriate reference value.
(TIF)

## Acknowledgments

The Authors would like to acknowledge all members of the Reproductive Health and Biotechnology Laboratory (University of Guelph), especially the Lab manager Monica Antenos and research technicians Elizabeth St. John and Allison MacKay for providing technical training and assistance. A special thank you to Dr. Gabriela Mastromonaco for her input in this project and for the stimulating discussions. Lastly, thank you to Cargill Meat Solutions (Guelph, ON) and Semex (Guelph, ON) for supplying the ovaries and the cryopreserved sperm, respectively.

## Author Contributions

**Conceptualization:** Laura A. Favetta.

**Data curation:** Alexander G. Kuzma-Hunt.

**Formal analysis:** Alexander G. Kuzma-Hunt, Reem Sabry, Ola S. Davis, Vivien B. Truong.

**Funding acquisition:** Laura A. Favetta.

**Investigation:** Alexander G. Kuzma-Hunt, Reem Sabry, Vivien B. Truong.

**Methodology:** Alexander G. Kuzma-Hunt, Reem Sabry, Ola S. Davis, Laura A. Favetta.

**Project administration:** Laura A. Favetta.

**Resources:** Jibran Y. Khokhar, Laura A. Favetta.

**Supervision:** Laura A. Favetta.

**Writing – original draft:** Alexander G. Kuzma-Hunt.

**Writing – review & editing:** Alexander G. Kuzma-Hunt, Reem Sabry, Jibran Y. Khokhar, Laura A. Favetta.

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
