## [Decision Letter · Decision Letter 0]

31 Oct 2023

PONE-D-23-30079THC and sperm: impact on fertilization capability, pre-implantation in vitro development and epigenetic modificationsPLOS ONE

Dear Dr. Favetta,

Thank you for submitting your manuscript to PLOS ONE. After a thorough review, we think it has merit, but could be improved if you followed the reviewers' advice and answered their questions to the best of your ability. Therefore, we invite you to submit a revised version of the manuscript that addresses the minor points raised during the review process.

Please submit your revised manuscript by Dec 15 2023 11:59PM. Please include the following items when submitting your revised manuscript:A rebuttal letter that responds to each point raised by the academic editor and reviewer(s). You should upload this letter as a separate file labeled 'Response to Reviewers'.A marked-up copy of your manuscript that highlights changes made to the original version. You should upload this as a separate file labeled 'Revised Manuscript with Track Changes'.An unmarked version of your revised paper without tracked changes. You should upload this as a separate file labeled 'Manuscript'.If applicable, we recommend that you deposit your laboratory protocols in protocols.io to enhance the reproducibility of your results. Protocols.io assigns your protocol its own identifier (DOI) so that it can be cited independently in the future. For instructions see: https://journals.plos.org/plosone/s/submission-guidelines#loc-laboratory-protocols. Additionally, PLOS ONE offers an option for publishing peer-reviewed Lab Protocol articles, which describe protocols hosted on protocols.io. Read more information on sharing protocols at https://plos.org/protocols?utm_medium=editorial-email&utm_source=authorletters&utm_campaign=protocols.

We look forward to receiving your revised manuscript.

Kind regards,

Joël R Drevet, Ph.D.

Academic Editor

PLOS ONE

2. We note that Figure 1, 2, 3 and 5 in your submission contain copyrighted images. All PLOS content is published under the Creative Commons Attribution License (CC BY 4.0), which means that the manuscript, images, and Supporting Information files will be freely available online, and any third party is permitted to access, download, copy, distribute, and use these materials in any way, even commercially, with proper attribution. For more information, see our copyright guidelines: http://journals.plos.org/plosone/s/licenses-and-copyright.

a. You may seek permission from the original copyright holder of Figure 1, 2, 3 and 5 to publish the content specifically under the CC BY 4.0 license. 

Additional Editor Comments:

Three reviewers with good expertise in the field evaluated this manuscript rather positively. They suggested minor revisions that could further improve this already good contribution. The authors are encouraged to follow these comments and advice and to respond to the various queries.

Reviewers' comments:

Reviewer's Responses to Questions

**Comments to the Author**

1. Is the manuscript technically sound, and do the data support the conclusions?

Reviewer #1: Yes

Reviewer #2: Yes

Reviewer #3: Yes

2. Has the statistical analysis been performed appropriately and rigorously? 

Reviewer #1: Yes

Reviewer #2: Yes

Reviewer #3: Yes

3. Have the authors made all data underlying the findings in their manuscript fully available?

Reviewer #1: Yes

Reviewer #2: Yes

Reviewer #3: Yes

4. Is the manuscript presented in an intelligible fashion and written in standard English?

Reviewer #1: Yes

Reviewer #2: Yes

Reviewer #3: Yes

5. Review Comments to the Author

Reviewer #1: I am sincerely appreciative of the opportunity I've been given to assess the manuscript authored by Laura A. Favetta and her colleagues, which is currently under consideration for publication in PLOS ONE. The primary objective of this research is to investigate the effects of in vitro exposure to THC on both the physical and functional aspects of sperm, as well as its inherent functions, particularly its role in embryo development subsequent to IVF. On the whole, I find the manuscript to be exceptionally well-structured and proficiently written. The data analysis is executed with precision, and the experimental methods are clearly explained, offering valuable insights into the impact of THC on male reproductive function. The Discussion section, while extensive, effectively delves into the physiological implications of the study's findings. It provides substantial evidence to support these implications and also acknowledges and addresses any limitations. Notably, the use of bovine sperm and oocytes represents a significant limitation in this study, but it has been thoroughly elucidated and acknowledged. Consequently, I recommend that the manuscript, in its current form, be accepted for publication.

Reviewer #2: Dear Authors,

The manuscript PONE-D-23-30079, titled "THC and sperm: impact on fertilization capability, pre-implantation in vitro development and epigenetic modifications," presents a compelling investigation into the effects of THC exposure on both morpho-physiological and intrinsic sperm functions. The study's findings provide a comprehensive understanding of how such exposure affects embryo development following in vitro fertilization. The manuscript is an original research article that uses a suitable methodology and is well-written in English. Moreover, it meets all adequate standards for the ethics of preclinical research and data availability, making it an impeccable report. Thus, I recommend this manuscript for publication in this journal. Nonetheless, this paper needs some prior minor modifications:

1) To improve your Introduction, I recommend that, in the sentence where you report the environmental and lifestyle factors (page 9, lines 74-7), you read and include the following article that discusses the association of cannabis among other several habits in male infertility (DOI: 10.3389/frph.2022.820451);

2) What are the references to the statement on page 10, lines 94-95? Are they the same as in the previous sentence (references 25 and 26)? Make this clear in the text;

3) Also in the Introduction, please include and explain the antagonist role of THC at CB receptors, as you reported on page 16, line 222;

4) I suggest the authors remove the sentence on page 13, lines 158-60, where they express the results of their research in the Introduction. My opinion is that it is better placed at the beginning of the discussion section if they want to maintain this information;

5) Please add the reference to the ethical principles for animal studies of the Canadian Council on Animal Care (page 13, line 166);

6) In the statistical analysis, specify the software versions and specifications, as usual.

Reviewer #3: Dear Authors, this manuscript is timely and appropriate. Congratulations on the initiative to provide the scientific and the public in general with additional valuable information on the matter of Marijuana and THC.

To further expedite the manuscript, please provide a few corrections and explanations:

Line 69: Male infertility accounts for approximately 20% of cases and contributes to another 50%. If that were the case, 20+50% would contribute to the male side of 70% overall! Did you mean to say that 30-35% is solely the male component, plus 20% of the couple's problem combined and the total contribution of the male component to the overall infertility is around 50%? Please correct.

Lines 147 - 148: Could you please provide further information on the choice of bovine sperm? We know that humans have the lowest sperm quality among mammals (the only exception for the Orangotangos) and that bovine sperm has a better quality as males have been selected through the years. So, the question is: Is there an ideal animal model to study bad-quality sperm? Another question is: if THC can affect a species with high-quality sperm, what to expect for humans? Even worst! Could you please add insights into the discussion?

Line 176: sperm was obtained through cryopreserved samples. It is known that cryopreservation can affect overall sperm quality. Both processes, cryopreservation and, thing can potentially have lethal or sublethal damage to the spermatozoa. Why have you not performed a group of fresh semen samples? Or, in the bovine model, is it not necessary to use fresh sperm?

Line 258: Please substitute 18 for Eighteen.

Dear authors, it may sound challenging to extrapolate the low-THC and high-THC concentrations. However, it would be nice if you could assume how many marijuana cigars these concentrations would mean in the human setting. We know that a marijuana cigar has around 0.3 grams of the Cannabis plant and that the THC concentrations vary from 9 to 30%. However, it can reach as much as 40 to 50% THC concentrations, depending on the plant's variety and breeding conditions. In the discussion section, if you believe that you could translate this research into practical information for users who use it rarely and for those who smoke regularly, believe it would be helpful in the clinical setting.

Discussion:

Line: 461. I believe that you have to include in the first phase: cryopreserved and thawed sperm.

It may sound too much or even unrealistic, but do you think it possible to include a figure with potential mechanistic and physiopathological pathways proposed here? If not, it is entirely understandable. Again, congratulations on the excellent piece of work.

6. PLOS authors have the option to publish the peer review history of their article (what does this mean?). If published, this will include your full peer review and any attached files.

Reviewer #1: No

Reviewer #2: No

Reviewer #3: **Yes: **Jorge Hallak, M.D.;Ph.D.

---

## [Author Response · Author response to Decision Letter 0]

10 Dec 2023

December 10th, 2023

Response to Reviewers PONE-D-23-30079

PLOS ONE

Dear Managing Editor/s,

We would like to thank the reviewers for their time, expertise, and comments on the manuscript. The authors agree with the concerns that were presented by the reviewers and have reformulated the manuscript to address these minor flaws. The authors have reviewed and revised the manuscript to address every concern presented by the reviewers (All additions/changes in the manuscript will appear as highlighted in yellow). We do believe that the manuscript is now substantially improved and deserving publication in PLOS ONE. 

Individual comments and authors’ responses to each comment are below:

Editor Comments/Responses:

Dear Editor, 

 We would like clarifications regarding the statement: “We note that Figure 1, 2, 3 and 5 in your submission contain copyrighted images.”

Figures 1, 2, 3 are the output of our experiments using flow-cytometry and we published these types of results multiple times in several journals and never had to provide copyright permission as they are not copyrighted, see some examples of recent publications below with similar flowcytometry outputs published.

• Davis OS., Truong VB., Hickey KD. and Favetta LA. (2023) Quality of fresh and cryopreserved bovine sperm is reduced by BPA and BPF exposure. Reprod Fertil. 2023 Sep 1:RAF-23-0018. doi: 10.1530/RAF-23-0018. Online ahead of print.

• Sabry R., Williams M., LaMarre J. and Favetta LA. (2023) Granulosa cells undergo BPA-induced apoptosis in a miR-21-independent manner. Exp Cell Res 427(1), 113574 doi: 10.1016/j.yexcr.2023.113574. 

• Truong VB., Davis OS., Gracey J., Neal MS., Khokhar JY. and Favetta LA. (2023) Sperm capacitation and transcripts levels are altered by in vitro THC exposure. BMC Molecular and Cell Biology. Feb 23;24(1):6. doi: 10.1186/s12860-023-00468-3.

• Dufour J., Sabry R., Khokhar JY. and Favetta LA. (2022) Delta-9 tetrahydrocannabinol (THC) effects on the cortisol stress response in bovine granulosa cells. Toxicol In Vitro. Dec 31:105549. doi: 10.1016/j.tiv.2022.105549. 

• Sabry R., Williams M., Werry N., LaMarre J. and Favetta LA. (2022) BPA decreases PDCD4 in bovine granulosa cells independently of miR-21 inhibition. International Journal of Molecular Sciences 23,8276. https://doi.org/ 10.3390/ijms23158276

Please, advise on what to do next, we could remove the outputs and leave only the graphs, but we would rather not if possible.

Figure 5 – these are a representation of images that we directly produced, therefore we cannot understand which copyright they would be under, as they are our original results.

Reviewer 1 Comments/Responses:

Dear Reviewer, 

Thank you for your positive feedback on our manuscript and your approval for publication. We appreciate your time and effort in reviewing our work. We are very excited about this work as well and cannot wait to see it published.

Reviewer 2 Comments/Responses:

Dear Reviewer, 

Thank you for your thoughtful feedback on our manuscript. We appreciate your time and effort in reviewing our work. We have carefully considered your comments and we addressed each point raised. 

1. To improve your Introduction, I recommend that, in the sentence where you report the environmental and lifestyle factors (page 9, lines 74-7), you read and include the following article that discusses the association of cannabis among other several habits in male infertility (DOI: 10.3389/frph.2022.820451)

Thank you for the suggestion, the article has been included and it is now reference # 8

2. What are the references to the statement on page 10, lines 94-95? Are they the same as in the previous sentence (references 25 and 26)? Make this clear in the text;

Thank you for your comment, clarity has been added.

3. Also in the Introduction, please include and explain the antagonist role of THC at CB receptors, as you reported on page 16, line 222.

Thank you for your comment, this is now been included in in the introduction in lines 101-103.

4. I suggest the authors remove the sentence on page 13, lines 158-60, where they express the results of their research in the Introduction. My opinion is that it is better placed at the beginning of the discussion section if they want to maintain this information;

Original lines 158-60 have been removed. As the authors present the main results already at the beginning of the discussion, we decided not to add these lines to the discussion as they might sound repetitive.

5) Please add the reference to the ethical principles for animal studies of the Canadian Council on Animal Care (page 13, line 166);

Reference has been added in line 167 and it is numbered 91

6) In the statistical analysis, specify the software versions and specifications, as usual.

Software version and specifications added in line 313: GraphPad Prism 8 (Version 8.4.3) and SPSS statistics software (Version 28.0.1.1) were used …

Reviewer 3 Comments/Responses:

Dear Reviewer, 

Thank you for your thoughtful feedback on our manuscript. We appreciate your time and effort in reviewing our work. We have carefully considered your comments and we addressed each point raised. 

Line 69: Male infertility accounts for approximately 20% of cases and contributes to another 50%. If that were the case, 20+50% would contribute to the male side of 70% overall! Did you mean to say that 30-35% is solely the male component, plus 20% of the couple's problem combined and the total contribution of the male component to the overall infertility is around 50%? Please correct.

Thank you for your comment, you are correct in your interpretation, that is now been clarified/correct in lines 70-71.

Lines 147 - 148: Could you please provide further information on the choice of bovine sperm? We know that humans have the lowest sperm quality among mammals (the only exception for the Orangotangos) and that bovine sperm has a better quality as males have been selected through the years. So, the question is: Is there an ideal animal model to study bad-quality sperm? Another question is: if THC can affect a species with high-quality sperm, what to expect for humans? Even worst! Could you please add insights into the discussion?

Thanks for your point, we fully agree with your comments and we already tried to address them in lines 715-720 of the original discussion. More insights have now been added in lines 720-728 of the revised manuscript.

Line 176: sperm was obtained through cryopreserved samples. It is known that cryopreservation can affect overall sperm quality. Both processes, cryopreservation and, thing can potentially have lethal or sublethal damage to the spermatozoa. Why have you not performed a group of fresh semen samples? Or, in the bovine model, is it not necessary to use fresh sperm?

We fully agree with the Reviewer’s comments and we now added a few lines addressing this in the discussion (lines 722-728). Experiments on fresh sperm were not performed for lack of availability of fresh sperm samples. In addition, performing the embryo production with fresh sperm would be extremely challenging, as we do not have fresh bovine sperm on site and the timing of collection and transport to the lab would not match with the timing of ovaries collection, transport to the lab from the slaughterhouse and oocytes retrieval, adding even more layers of confounding variables (e.g. time between collection and use). This is a reason why cryopreserved sperm is often used in research due to the availability and ease of storing a high number of samples.

Line 258: Please substitute 18 for Eighteen.

Done

Dear authors, it may sound challenging to extrapolate the low-THC and high-THC concentrations. However, it would be nice if you could assume how many marijuana cigars these concentrations would mean in the human setting. We know that a marijuana cigar has around 0.3 grams of the Cannabis plant and that the THC concentrations vary from 9 to 30%. However, it can reach as much as 40 to 50% THC concentrations, depending on the plant's variety and breeding conditions. In the discussion section, if you believe that you could translate this research into practical information for users who use it rarely and for those who smoke regularly, believe it would be helpful in the clinical setting.

We thank the reviewer for their comments and the encouragement to attempt to translate our results to a clinical setting. Unfortunately, we do believe that because of many reasons, including the main that the reviewer mentions as well (THC concentration varying between 9 and 50% and depending of the plant’s variety), it would not be realistic to translate these in vitro experiments to an in vivo setting and might give not correct clinical information. While we fully stand behind the in vitro doses used in this article as they do reflect a window of concentrations found in human seminal plasma based on different range of use (Whan et al., 2006), trying to further narrow this down to number of cigars/cigarettes smoked per day or type/amount/number of edibles and frequency is unfortunately beyond the scope of this study. 

Discussion:

Line: 461. I believe that you have to include in the first phase: cryopreserved and thawed sperm. It may sound too much or even unrealistic, but do you think it possible to include a figure with potential mechanistic and physiopathological pathways proposed here? If not, it is entirely understandable. Again, congratulations on the excellent piece of work.

 We agree that we should state right at the beginning of the discussion that sperm was cryopreserved and thawed, this is now clearly stated in line 463.

We are actually working on a review on this topic and trying to include diagram representations of possible mechanistic pathways (not much out there, mostly speculations based on results). We value the reviewer’s suggestion, but finds it more fitting in the working review paper.

We believe we have addressed all the comments/inquiries at the best of our ability and believe the manuscript is substantially improved and worth of publication

Do not hesitate to contact us, if you require additional information

Kind Regards,

Laura Favetta

Laura Favetta, Ph.D.

Associate Professor

Faculty Advisor-MBS program in Applied Reproductive Biotechnologies

Department of Biomedical Sciences, University of Guelph

Guelph, ON N1G 2W1

Ph: (519) 824-4120 ext. 56212

E-mail: lfavetta@uoguelph.ca

Rm. 3621

---

## [Decision Letter · Decision Letter 1]

30 Jan 2024

THC and sperm: impact on fertilization capability, pre-implantation in vitro development and epigenetic modifications

PONE-D-23-30079R1

Dear Dr. Laura Favetta,

We’re pleased to inform you that your manuscript has been judged scientifically suitable for publication and will be formally accepted for publication once it meets all outstanding technical requirements.

Kind regards,

Joël R Drevet, Ph.D.

Academic Editor

PLOS ONE

Additional Editor Comments (optional):

Thank you for the changes made to your manuscript in response to the requests/requirements of the reviewers.

Reviewers' comments:

Reviewer's Responses to Questions

**Comments to the Author**

1. If the authors have adequately addressed your comments raised in a previous round of review and you feel that this manuscript is now acceptable for publication, you may indicate that here to bypass the “Comments to the Author” section, enter your conflict of interest statement in the “Confidential to Editor” section, and submit your "Accept" recommendation.

Reviewer #2: All comments have been addressed

2. Is the manuscript technically sound, and do the data support the conclusions?

Reviewer #2: Yes

3. Has the statistical analysis been performed appropriately and rigorously? 

Reviewer #2: Yes

4. Have the authors made all data underlying the findings in their manuscript fully available?

Reviewer #2: Yes

5. Is the manuscript presented in an intelligible fashion and written in standard English?

Reviewer #2: Yes

6. Review Comments to the Author

Reviewer #2: Dear authors,

You have adequately addressed your comments raised in a previous round of review and you feel that this manuscript is now acceptable for publication.

7. PLOS authors have the option to publish the peer review history of their article (what does this mean?). If published, this will include your full peer review and any attached files.

Reviewer #2: No

---

## [Editor Report · Acceptance letter]

1 Mar 2024

PONE-D-23-30079R1 

PLOS ONE

Dear Dr. Favetta, 

I'm pleased to inform you that your manuscript has been deemed suitable for publication in PLOS ONE. Congratulations! Your manuscript is now being handed over to our production team.

Kind regards, 

on behalf of

Prof. Joël R Drevet 

Academic Editor

PLOS ONE